# Performance of Modular-Reinforced Soil-Retaining Walls for an Intercity Railway during Service

**Xiaoyong Liang [1,2], Jing Jin [2], Guangqing Yang [3,4,*], Xizhao Wang [2], Quansheng Zhao [2] and Yitao Zhou [5]**

1 School of Traffic and Transportation, Shijiazhuang Tiedao University, Shijiazhuang 050043, China; liaxiayon@hebust.edu.cn

2 School of Civil Engineering, Hebei University of Science and Technology, Shijiazhuang 050018, China; jinjing@hebust.edu.cn (J.J.); wxizhao@hebust.edu.cn (X.W.); zhao_quansheng@hebust.edu.cn (Q.Z.)

3 State Key Laboratory of Mechanical Behavior and System Safety of Traffic Engineering Structures, Shijiazhuang Tiedao University, Shijiazhuang 050043, China

4 School of Civil Engineering, Shijiazhuang Tiedao University, Shijiazhuang 050043, China

5 Hebei Key Laboratory of Geotechnical Engineering Safety and Deformation Control, Hebei University of Water Resources and Electric Engineering, Cangzhou 061001, China; zhouytwr@163.com

* Correspondence: yanggq@stdu.edu.cn

**Abstract:** In order to investigate the mechanical behavior of reinforced soil-retaining walls during service, this paper carried out a long-term remote observation test for 6 years on the modular-reinforced soil-retaining wall of the Qingrong intercity railway in eastern China's Shandong Province. During the construction period, earth pressure boxes, flexible displacement meters, settlement pipes and displacement meters were buried to observe the soil pressure, reinforcement strain, horizontal displacement and settlement of the reinforced earth-retaining wall, respectively, for a long time; then, the results were analyzed to summarize its variation law. The results show that the reinforced earth-retaining wall was stable after one year of construction. It was determined that the strain of reinforcement in each layer decreased with time, culminating in a value of less than 0.88 percent during the 6th year. The maximum horizontal displacement of the wall was 11.43 mm and the maximum settlement of the wall top was 46.77 mm, which were 0.15% and 0.60% of the wall height, respectively. These research results can be applied to the construction and design of reinforced soil-retaining walls in high-speed railways. The effects of the elastic modulus of filler, the tensile modulus of reinforcement and the reinforcement length on the characteristics of the retaining wall were analyzed in the numerical simulation with PLAXIS2D. The results and analysis show: the elastic modulus of filler and reinforcement length have a significant effect on the horizontal displacement of the retaining wall. The results of this experiment can be referenced for engineering projects.

**Keywords:** modular-reinforced soil-retaining wall; vertical earth pressure; lateral earth pressure; geogrid strain; horizontal displacement

## 1. Introduction

A reinforced soil structure can effectively improve the strength of the soil and reduce the deformation of the soil. As one of its main forms, the reinforced soil-retaining wall has been extensively applied in the engineering field [1–5]. According to the wall form, reinforced soil-retaining walls can be divided into wrap-reinforced soil-retaining walls [6], modular-reinforced soil-retaining walls [7], rigid panel-reinforced soil-retaining walls [8] and gabion-reinforced soil-retaining walls [9] etc.

In recent years, topics related to reinforced soil-retaining walls have been studied by scholars at home and abroad to meet the needs of engineering developments. Tatsuoka [10] briefly analyzed the structural characteristics of modular, backpacked and integral panel-reinforced retaining walls. Balakrishnan [11] investigated the effect of different stiffnesses on the horizontal displacement of reinforced soil-retaining walls, the wall roof settlement

and the maximum peak strain of the grid by centrifugal model test. Razeghi [12] simulated the rise of groundwater and analyzed the influence of existing and soil-less synthetic material drainage layers on the displacement of reinforced soil-retaining walls and the internal pore water pressure change by centrifugation. Mehrjardi [13] considered the size effect of fill particles, geogrid aperture and size effect of loading plates on reinforced soil-retaining walls; the response of the retaining wall model was evaluated by the applied load and surface settlement, so as to further understand the characteristics of reinforced soil-retaining walls. To investigate the influence of the permeability of soil filling on the structural performance of the retaining wall, Portelinha [14] established a full-size reinforced soil-retaining wall using the irrigation system to simulate the rainfall process; the author then measured the changes in soil filling volume moisture content, substrate suction, wall displacement and grille strain. Bathurst [15,16] studied the effect of wall stiffness on the structural performance of modular- and envelope-retaining walls. Ehrlich et al. [17] studied the effect of filler on the performance of the retaining wall under different compaction degrees. Ehrlich et al. [18] used models to test the modular- and wrap-reinforced retaining wall to study the impact of wall stiffness and wall toe constraints on the performance of soil walls reinforced with geosynthetic materials. Xiao et al. [19] studied the factors affecting the load-carrying capacity of the reinforced soil-retaining wall. Yazdandoust [20] performed the vibration table test of modular panel-reinforced soil-retaining walls with different band lengths to study the effect of band length on the deformation mode. Xu et al. [21,22] studied the dynamic response of the overall rigid panel-reinforced retaining wall by investigating the phase difference between reinforced areas and unreinforced areas through the vibration platform test and proposed an analytical method to determine the yield acceleration and lateral displacement of the reinforced soil-retaining wall. Wang et al. [23] investigated the structural properties of modular panel-reinforced soil-retaining walls with deformation buffers through model tests and numerical simulations. Yang et al. [24] developed a numerical model of the double-sided reinforced soil-retaining wall to investigate its damage patterns under transverse loads like floods, tsunamis, mudslides and avalanches. Lu Liang et al. [25] developed the prestressed reinforcement soil-retaining wall structure and proposed the calculation method of the corresponding lateral displacement and lateral moving soil pressure combined with the vibration platform model test. Yang et al. [26] investigated the deformation and strain of reinforced soil walls through the FEM method. Yoo et al. [27,28] analyzed the wall stress, wall deformation and reinforcement strain during the construction period and after completion through the field full-scale test and a numerical simulation of the two-stage reinforced earth-retaining wall. Liu et al. [29] studied the seismic response of reinforced earth-retaining walls through numerical simulation.

There were few research tests conducted on the site, especially as the field data surrounding the modular-reinforced earth barrier wall in the high-speed railway was more valuable. In this paper, the mechanical behavior and deformation law of the structure during operation were studied, combined with the field test of the Qingrong passenger-dedicated railway. The effects of the elastic modulus of filler, the tensile modulus of reinforcement and the reinforcement length on the characteristics of the soil-retaining wall were analyzed in the numerical simulation with PLAXIS. The modular-reinforced soil-retaining wall has many advantages, such as flexible design, high construction efficiency, less area, cost-efficiency and good flexibility. It is widely used in the construction of high-speed railways, which is bound to bring considerable social and economic benefits.

## 2. Remote Monitoring during Service

### 2.1. Project Overview

The Qing (Dao) -Rong (Cheng) intercity railway, located in the eastern Shandong Province, was the first regional high-speed railway with a designed speed of 250 km/h. The line starts from Qingdao Station in the south and goes to Rongcheng Station in the east; it started construction on 10 October 2010 and was put into service from Jimo to Rongcheng on 28 December 2014. There were 15 stations in total with a total length of 316 km. Based

on the modular-reinforced soil-retaining wall of the Qingrong Intercity Railway, section DK39 + 610 was selected, as shown in Figure 1. An experimental study using remote observation was conducted during the service period of the structure, and remote monitoring started after the completion of the construction of the reinforced soil-retaining wall.

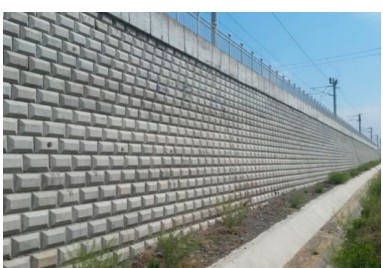

**Figure 1.** Modular-reinforced earth-retaining wall of Qingrong Intercity Railway.

### 2.2. Backfill Soil

The cohesion and internal friction angle of backfill were measured by laboratory directed shearing test, and the maximum dry density was determined by compaction test, as shown in Table 1. The gradation of soil was obtained through a screening method, and the curve was drawn, as shown in Figure 2. Comprehensive laboratory results show that backfill soil was unsaturated soil (uniform coefficient $C_u$ = 6.09, curvature coefficient $C_c$ = 1.13) and it was gravel soil group B packing.

**Table 1.** Physical and Mechanical Properties of Backfill.

| Maximum Dry Density /cm³ | Moisture Content /% | Optimum Moisture Content /% | Uniformity Coefficient /$C_u$ | Curvature Coefficient /$C_c$ | Cohesion /kPa | Friction Angle /° |
|---|---|---|---|---|---|---|
| 2.23 | 5.7 | 7.8 | 6.09 | 1.13 | 3.1 | 40 |

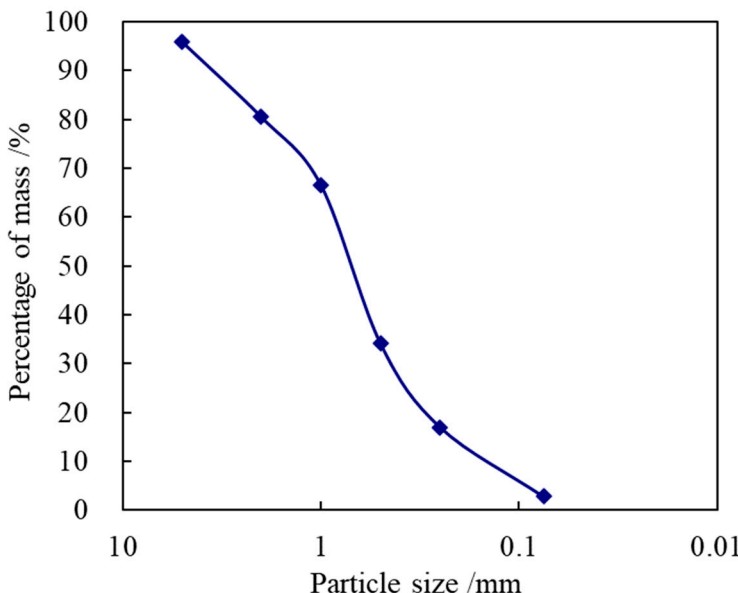

**Figure 2.** Curve of particle size distribution of the backfill.

### 2.3. Geogrid

The uniaxial geogrids composed of HDPE (EG130R) were used in the wall. A total of 23 layers of geogrids were laid as the foundation; the effective length of the geogrids was 8 m from the 1st layer to the 19th layer and 10 m from the 20th layer to the 23rd layer, with a vertical distance of 0.3 m. The mechanical properties of the geogrid measured in the laboratory are shown in Table 2.

**Table 2.** Mechanical property indexes of geogrid.

| Specification | Tensile Strength/(kN/m) | Tensile Strength of 2% Strain/(kN/m) | Tensile Strength of 5% Strain/(kN/m) | Peak Strain/% |
|---|---|---|---|---|
| EG130R | 130 | 36.5 | 72 | 11.28 |

*2.4. Structural Type and Instrument Layout of Reinforced Retaining Wall*

The height, width and thickness of the modular-reinforced retaining wall were 300 mm, 500 mm and 300 mm, respectively. The geogrid transverse ribs were pre-casted in the module, with at least one transverse rib reserved on the outer side to connect the geogrid to the exposed reinforcement of the module by connecting the rods, as shown in Figure 3. The mortar block stone built on the bedrock was used as the foundation of the whole reinforced soil-retaining wall.

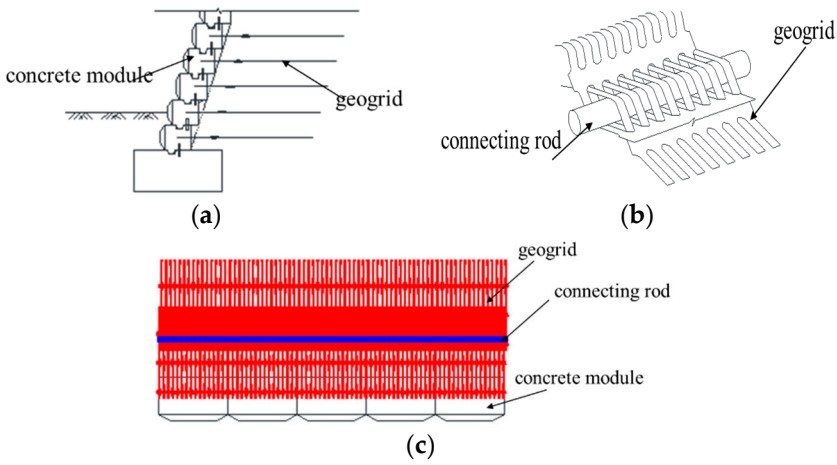

**Figure 3.** Connection mode between module and geogrid. (**a**) Embedded grid in module; (**b**) Connection detail; (**c**) Schematic diagram of module and grid connection.

Since the horizontal earth pressure of the wall was mainly borne by the backpack, the soil pressure cells were installed in the horizontal and vertical directions of the backpack to test the vertical earth pressure and lateral earth pressure. The flexible displacement meter was installed on the geogrid to test the deformation of the geogrid. The inclinometer tube (length of 12 m) was installed at 0.5 m from the inside of the capstone to the test wall displacement; the single point settlement meter was installed at 5.1 m from the inside of capstone to test settlement and buried to a depth of 12 m.

The retaining wall was monitored using wireless monitoring instruments. The test equipment includes horizontal and vertical earth pressure boxes, flexible displacement meters, single point displacement meter and inclinometer tubes; test element parameters are shown in Table 3. The DK39 + 610 section was monitored and the test sensors were buried in layers 1, 5, 10, 15 and 20. The section and sensor arrangements are shown in Figure 4. The main monitoring equipment is shown in Figure 5.

**Table 3.** Field test equipment parameters.

| Equipment Name | Equipment Type | Equipment Range | Equipment Number |
|---|---|---|---|
| Flexible displacement meter | JMDL-2405A | 50 mm | 40 |
| Earth pressure box | JMZX-5003A | 0.3 MPa | 68 |
| Single point displacement meter | JMDL-4710A | 100 mm | 2 |
| Inclinometer tube | JMZXX-3001 | — | 1 |

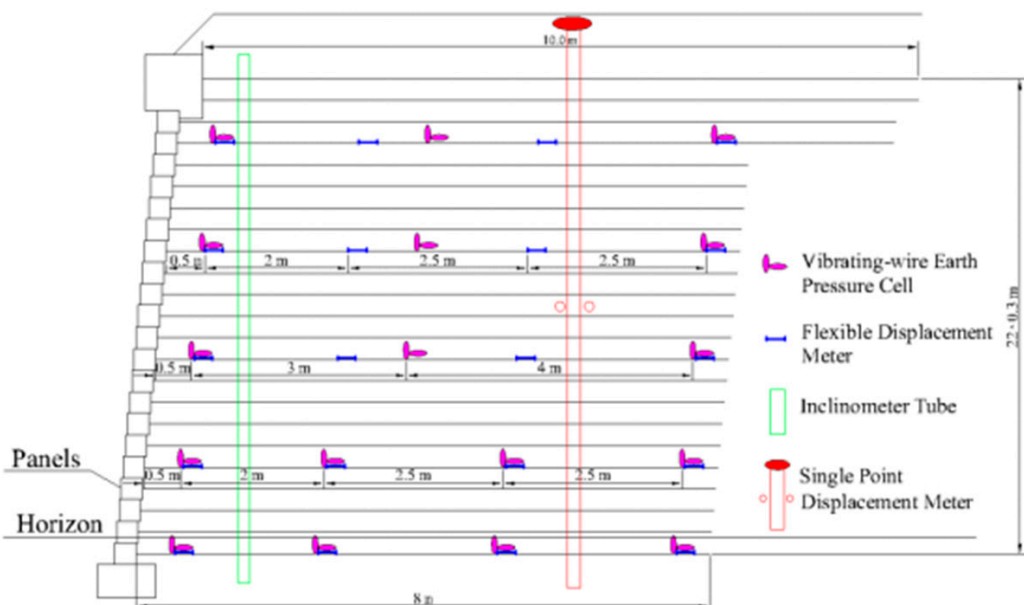

**Figure 4.** Arrangement of monitoring instruments.

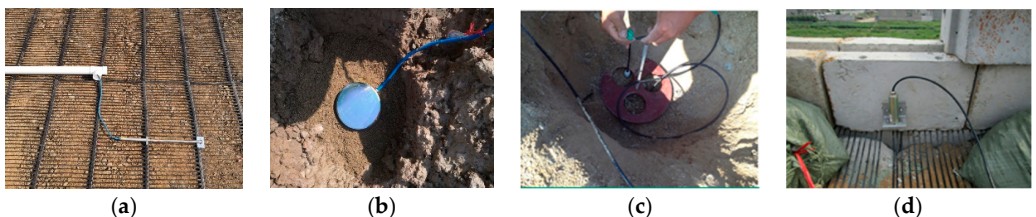

| (**a**) | (**b**) | (**c**) | (**d**) |

**Figure 5.** Monitoring instruments: (**a**) strain gauge; (**b**) earth pressure cell; (**c**) settlement meter; (**d**) displacement meter.

## 3. Monitoring Results and Analysis

### 3.1. Vertical Soil Pressure inside the Retaining Wall

During the service period, the vertical earth pressure distribution curve of each layer in the inside of the retaining wall over time are shown in Figure 6.

It can be seen that the vertical earth pressure near the retaining wall panel and the inside of the retaining wall at different heights showed a downward trend, and then this trend gradually stabilized as time went on. The main reasons were as follows: (1) Soil was consolidated with time and the vertical earth pressure tends to stabilize gradually. (2) The friction between the wall and the filler also affected the distribution of the vertical earth pressure. (3) The "net pocket" effect produced by the geogrid showed a downward trend in the earth pressure after construction. With time, the reinforced structure was gradually stabilized, while the earth pressure in the later stages remained basically unchanged.

The vertical earth pressure inside the retaining wall changes more steadily with greater distance from the top of the wall. The closer to the wall top, the more significant the change in vertical earth pressure. This was because: (1) The action range and position of the train load affects the distribution of the vertical earth pressure; (2) The tensile force of the lower tie bar plays a better role.

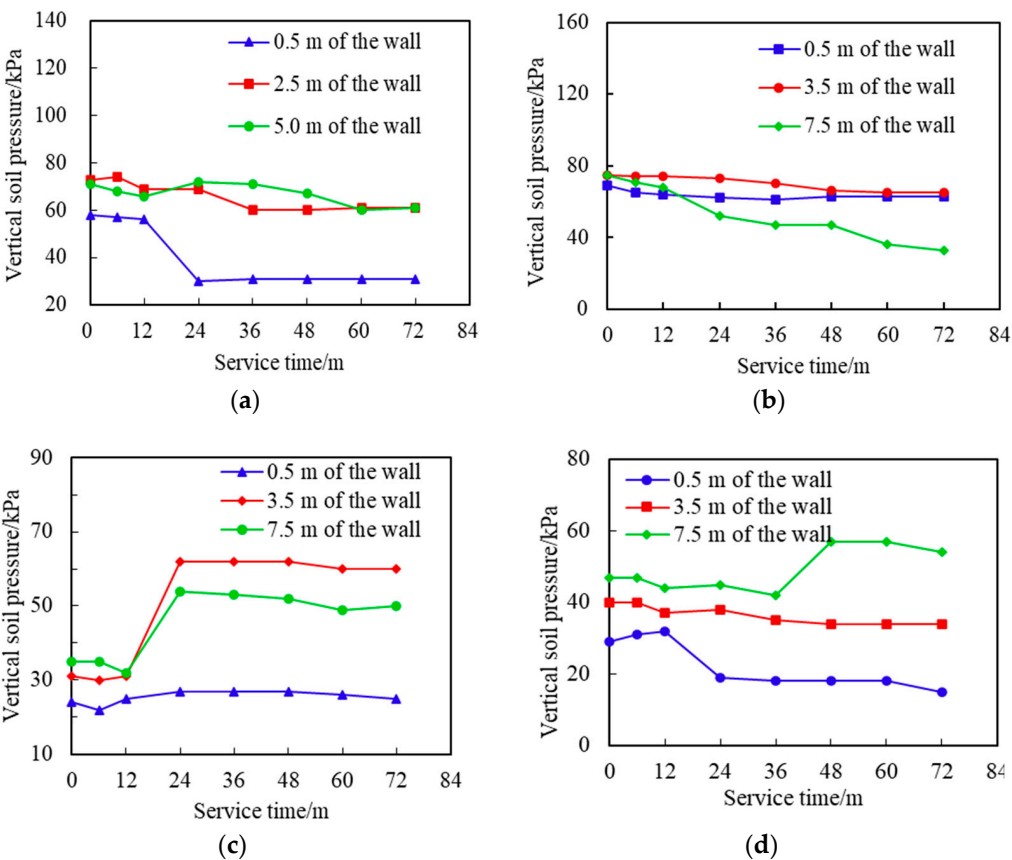

**Figure 6.** Vertical earth pressure in the retaining wall. (**a**) The 5th level; (**b**) The 10th level; (**c**) The 15th level; (**d**) The 20th level.

### 3.2. Horizontal Soil Pressure on the Back of Wall

Horizontal earth pressure on the back of the wall varies with the height of the wall, as presented in Figure 7, in which the horizontal earth pressure on the back of the wall first decreases and then increases with the change in wall height, and finally decreases with the nonlinear change trend. With the increase in service time, the data was relatively stable. Six years after completion, the maximum horizontal earth pressure of the reinforced earth-retaining wall panel was 6 kPa lower than that at completion and the pressure values changed very little. The main reason for this was that the horizontal displacement of the wall gradually increased and the foundation was deformed, resulting in the reduction of the vertical earth pressure and a subsequent reduction in the horizontal earth pressure.

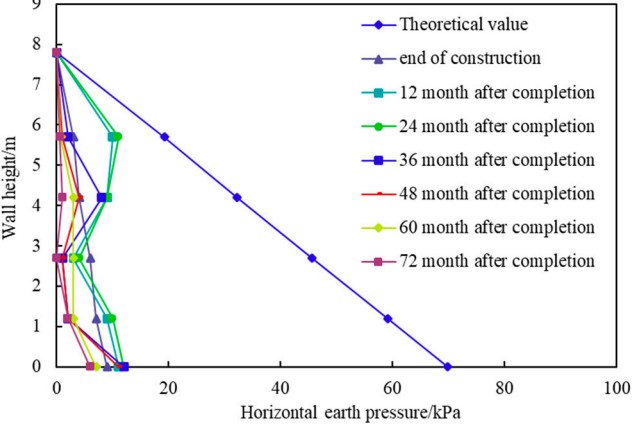

**Figure 7.** Horizontal earth pressure on the back of the wall.

By comparing the measured value on the back of wall with the theoretical value, the results showed that the measured value was less than the soil pressure value calculated according to the classical theory, and it was suggested that the theory could be used to calculate the horizontal soil pressure.

### 3.3. Reinforcement Strain

The curve of the geogrid strain at different layers after completion of the construction sections is shown in Figure 8. The geogrid strain of each layer showed a single peak distribution along the length of reinforcement. In these particular cases, double peaks of strain distribution appeared, with a peak near the wall caused by the lateral earth pressure of the filler and a peak value far away from the wall due to the joint action of soil weight and friction resistance between the filler and the geogrid. The strain of each layer basically showed a downward trend with the continuation of service time, and the decreasing range was less than 0.88 percent, indicating that the tension of the reinforcement was small at a current service period of 6 years.

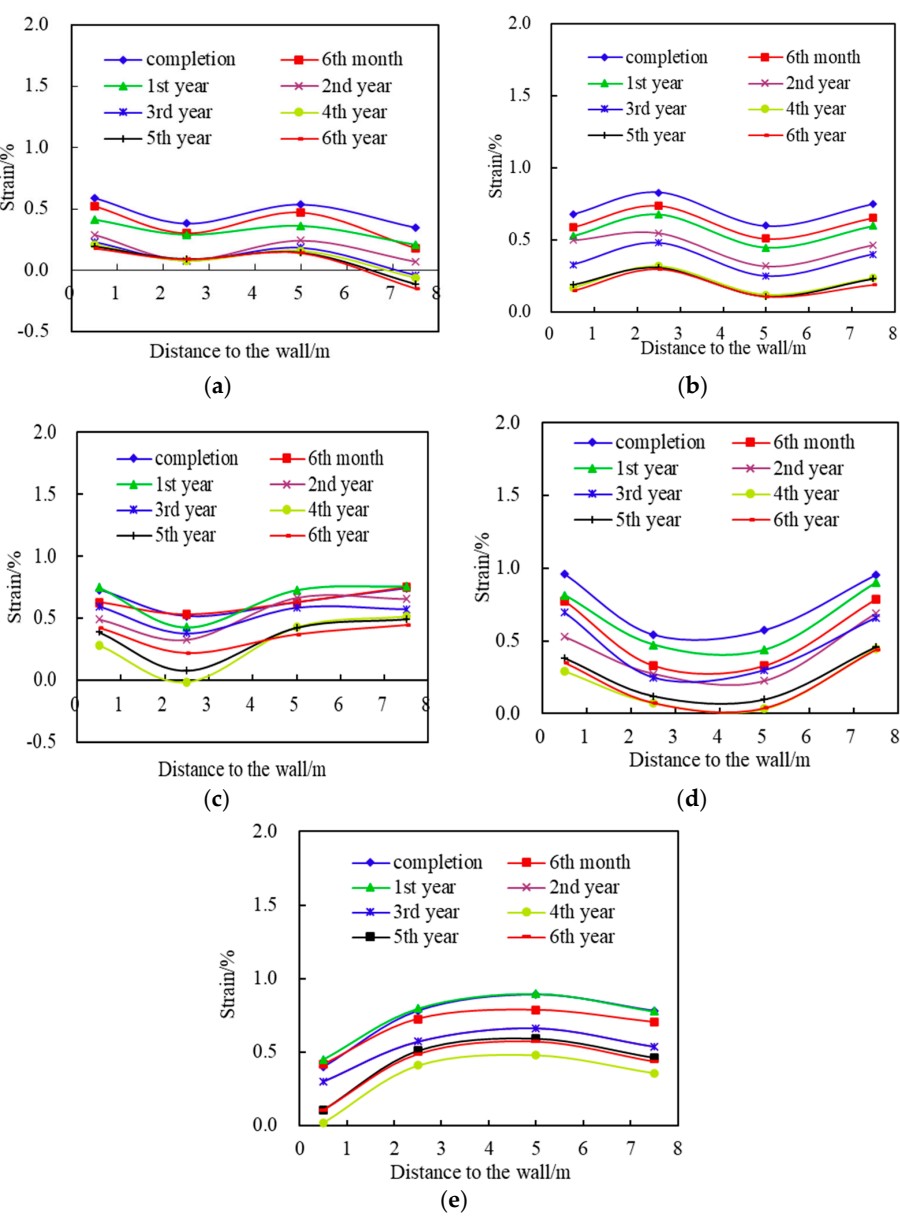

**Figure 8.** Geogrid strain during service time. (**a**) The 1st layer; (**b**) The 5th layer; (**c**) The 10th layer; (**d**) The 15th layer; (**e**) The 20th layer.

### 3.4. Horizontal Displacement of the Retaining Wall

The plot of the horizontal displacement vs. the service time is shown in Figure 9. The horizontal displacement of the wall gradually showed a stable trend over the duration of service time, with the maximum value of the horizontal displacement occurring at a height of 5 m. Primarily due to the inner edge of the load being 3.358 m away from the inner side of the cap stone, the maximum displacement appeared nearby due to the effect of the stress diffusion angle.

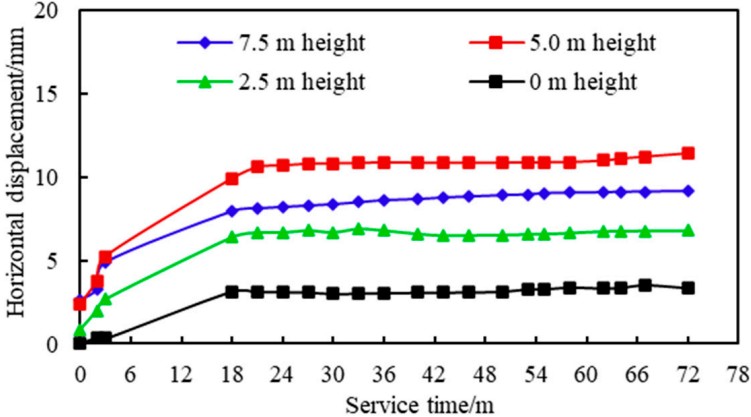

**Figure 9.** Horizontal displacement over the service time.

Within 6 years of completion, the maximum horizontal displacement of the wall was 11.43 mm, which was 15 percent of the wall height and far less than the limit of 1.67 percent specified by AASHTO of the United States.

### 3.5. Settlement of the Soil-Retaining Wall Top

Over the service time of six years, variation in the settlement of the retaining wall top was obtained through the single point displacement meter in the Qingrong intercity railway. It was designed with an intended speed of 250 km/h; if the structure was to reach a stable state, its post-construction settlement should not be greater than 10 cm and the settlement rate should not be greater than 3 cm per year, according to the Code for Design of High-speed Railway. The curve of the settlement over the service time is shown in Figure 10. The settlement of the wall top increased gradually, and the settlement rate gradually decreased with the extension of the service time; the maximum settlement of the wall top was 46.77 mm, which was 0.60 percent of the wall height and achieved the requirements for the post-construction settlement of a ballasted track road with a speed of 250 km/h.

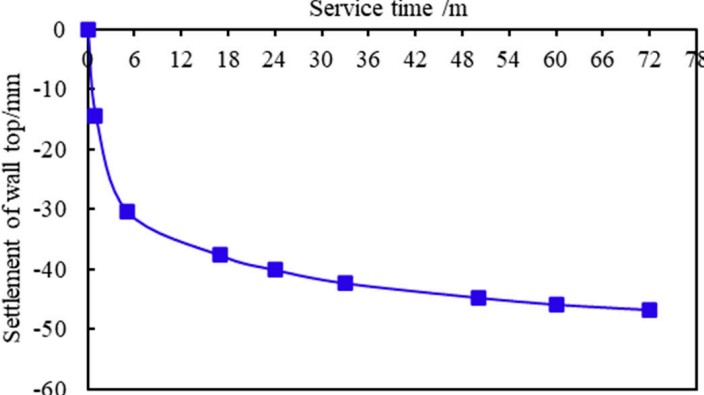

**Figure 10.** Settlement of the wall top over the service time.

The change in the settlement rate of the reinforced earth-retaining wall during the service time is shown in Table 4. The settlement rate was 3.6 cm/y in the first year, which was greater than the limit value in the specification; whereas from the second year to the sixth year, the settlement rate meets the specification requirements, which indicates that the structure reached a stable state one year after the completion of construction.

**Table 4.** Sedimentation rate over the service time.

| Service time/y | 1 | 2 | 3 | 4 | 5 | 6 |
|---|---|---|---|---|---|---|
| Settlement rate/(cm/y) | 3.6 | 0.41 | 0.29 | 0.17 | 0.11 | 0.09 |

Table 4 shows the table of the settlement rate change during the service of the reinforced soil-retaining wall. It can be seen that the settlement rate for the first year was 3.6 cm/year, which was greater than the limit in the specification; however, the remaining settlement rates met the specification requirements, which shows that the retaining wall reached a stable state one year after the completion of construction.

## 4. Constitutive Model and Material Parameters of Numerical Simulation

The service period of the reinforced earth-retaining wall was analyzed by PLAXIS. Due to the large length of the reinforced soil-retaining wall along the longitudinal direction, the longitudinal displacement can be ignored, which can be simulated by the plane strain model. In the numerical calculation, the 15-node triangular element was adopted for the constitutive relationship; the foundation and soil-retaining wall fill were regarded as elastic−plastic bodies and the Mohr Coulomb model was adopted. The wall panel adopted a 5-node beam element. The elastic modulus of the geogrid was 1825 MPa and the constitutive model adopted linear elasticity. The reinforcement was simulated using a geotextile element, and the interface coefficient of the reinforcement and soil was 0.67. The wall top and panel of the reinforced soil-retaining wall were set free from boundaries; the back of the retaining wall was subject to horizontal constraints and the base of the retaining wall was subject to horizontal and vertical constraints. The grid division of the retaining wall is shown in Figure 11 and the material characteristic parameters are shown in Table 5.

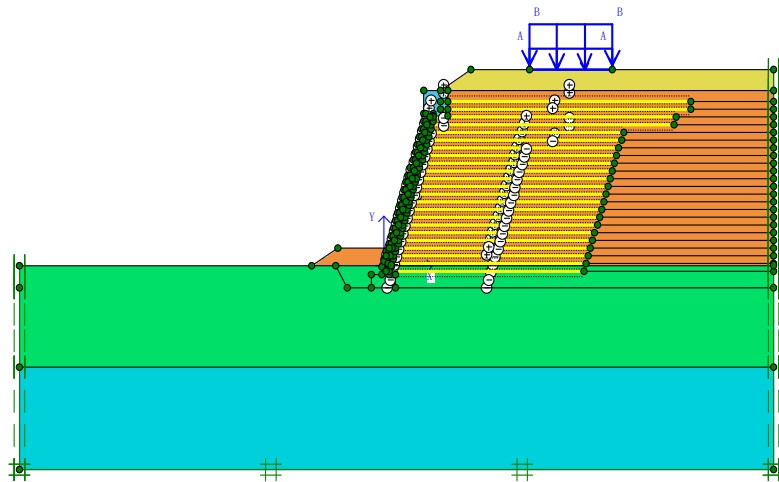

**Figure 11.** Grid division of retaining wall.

**Table 5.** Material parameters in numerical simulation.

| Material | Gravity /(kN·m$^{-3}$) | Elastic Modulus /MPa | Poisson Ratio | Cohesion /kPa | Internal Friction Angle/° |
|---|---|---|---|---|---|
| Filler | 20.0 | 50.0 | 0.3 | 6 | 35 |
| Foundation | 24.0 | $3.0 \times 10^4$ | 0.20 | — | — |

## 5. Influencing Factors on the Horizontal Deformation of Modular-Reinforced Earth-Retaining Wall

In order to ensure that the horizontal deformation of the reinforced earth-retaining wall met the engineering requirements, the influencing factors on the horizontal deformation of the reinforced earth-retaining wall were further studied. This paper mainly analyzed the influence of the filler elastic modulus, the reinforcement tensile modulus and the reinforcement length on the horizontal deformation of the wall.

### 5.1. Elastic Modulus of Filler

The filler was an important part of reinforced earth structure, which together with the geogrid and the module form the reinforced earth-retaining wall structure. In order to study the influence of the elastic modulus of the filler on the horizontal deformation of the retaining wall, the elastic moduli were taken as 20, 25, 30, 35, 40, 45 and 50 MPa, respectively. The distribution law of the horizontal displacement along the wall height at different times under different elastic moduli of filler is shown in Figure 12.

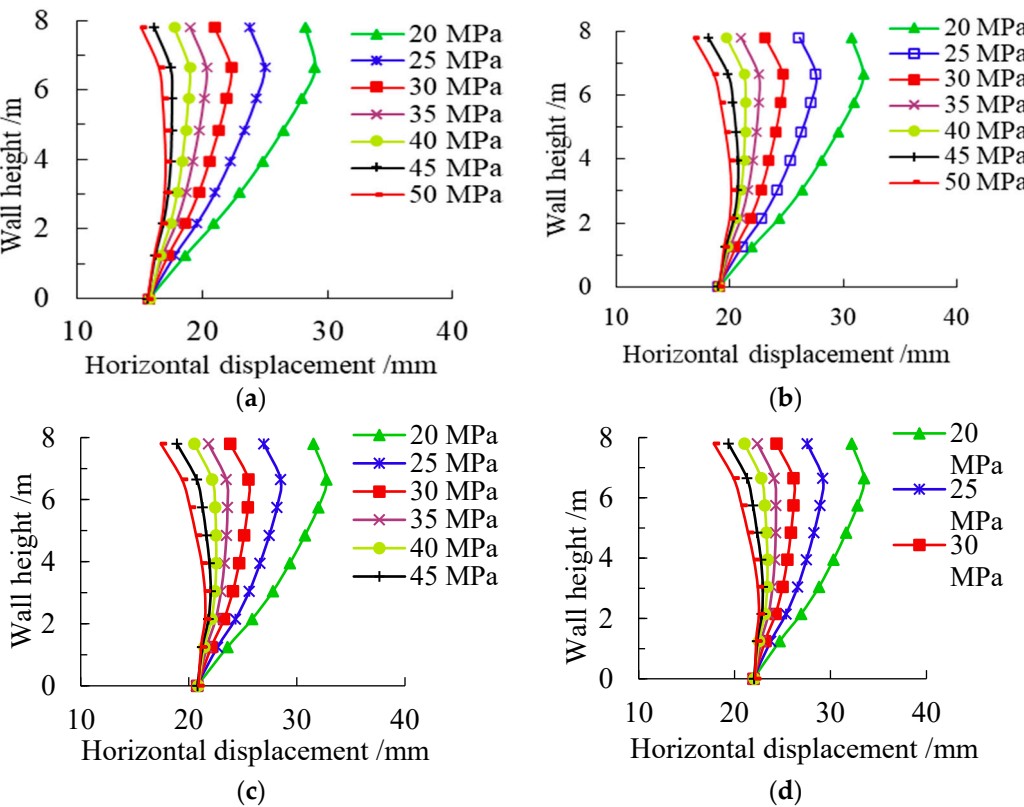

**Figure 12.** Influence of elastic modulus on horizontal displacement of the wall during service time. (**a**) The 5th layer; (**b**) The 10th layer; (**c**) The 15th layer; (**d**) The 20th layer.

It can be seen from the figure that the variation law of the horizontal displacement of the retaining wall was basically the same, the lateral displacement of the wall bottom changed little, the deformation of the middle and upper part was large and the distribution along the wall height was "belly shaped". The position of the maximum horizontal displacement gradually moved from the upper part to the middle part with the increase in the elastic modulus. To the middle of the wall, the change range showed a decreasing trend and gradually became stable. This was mainly because the larger the elastic modulus of filler, the stronger its integrity and the stronger its ability to resist deformation. After 5 years of service, the horizontal displacement changed greatly. The maximum horizontal displacement corresponding to the elastic modulus of different fillers was between

17.1–30 mm and the horizontal displacement of the wall decreased with the increase of the elastic modulus.

After 5, 10, 15 and 20 years of service, the maximum horizontal displacement was 29 mm, 31.8 mm, 32.73 mm and 33.48 mm, respectively. It can be seen that the change in horizontal displacement after construction decreased gradually as time went on.

### 5.2. Tensile Modulus of Reinforcement

In order to meet the deformation and stability requirements of the high-speed railway retaining wall, it was very important to select reinforcements with a reasonable tensile modulus. The influence on the horizontal deformation of the retaining wall is shown in Figure 13 when the tensile modulus was 500, 1000, 1825, 2000 and 4000 kN/m.

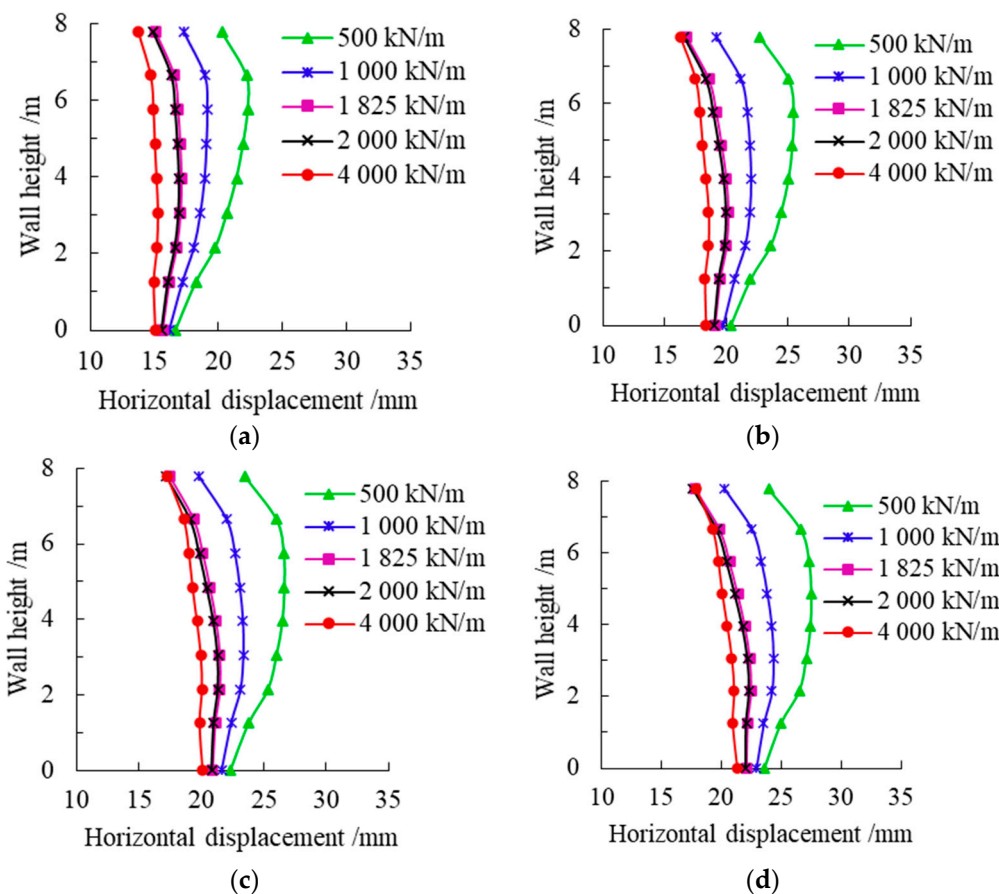

**Figure 13.** Influence of tensile modulus on horizontal displacement of the wall during service time. (**a**) The 5th layer; (**b**) The 10th layer; (**c**) The 15th layer; (**d**) The 20th layer.

It can be seen that the variation law of the horizontal displacement of the retaining wall was basically the same; the lateral displacement at the bottom of the wall did not change much, the deformation of the middle and upper part was large, and the distribution along the wall height was "belly shaped". With the increase of the geogrid tensile modulus, the horizontal displacement of the wall decreased gradually, and the reduction rate was smaller and smaller. When the tensile modulus of the geogrid increased from 500 kN/m to 1000 kN/m, the horizontal displacement of the wall decreased greatly. This was mainly because the deformation resistance of the geogrid played a key role in the horizontal displacement of the retaining wall. The larger the tensile modulus of the geogrid, the stronger its deformation resistance, which further limited the movement of soil particles. Therefore, horizontal displacement of the wall decreased with the decrease in the tensile modulus. After 5 years of service, the maximum horizontal displacement corresponding

to different geogrid tensile moduli was 15.3–22.3 mm. With the increase in the tensile modulus, the horizontal displacement of the wall decreased.

After 5, 10, 15 and 20 years of service, the maximum horizontal displacement was 22.31 mm, 25.49 mm, 26.6 mm and 27.35 mm, respectively. It can be seen that the change in horizontal displacement after construction decreased gradually as time went on.

### 5.3. Reinforcement Length

The influence of different geogrid lengths on horizontal displacement was analyzed by increasing or decreasing the proportion of the geogrid, as shown in Figure 14.

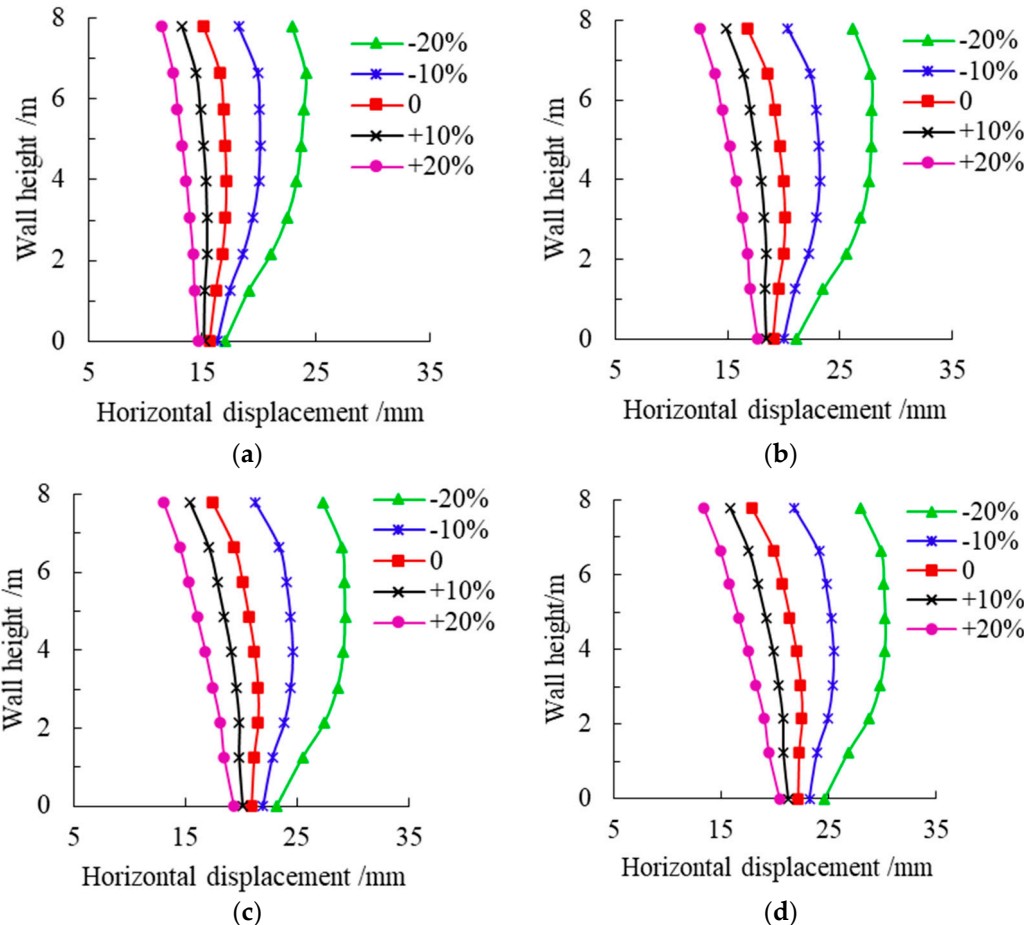

**Figure 14.** Influence of geogrid length on horizontal displacement of the wall during service time. (**a**) The 5th layer; (**b**) The 10th layer; (**c**) The15th layer; (**d**) The 20th layer.

It can be seen that variation law of horizontal displacement of the retaining wall was basically the same. The lateral displacement of the wall bottom changed little and the deformation of the middle and upper part was large. With the increase in geogrid length, the maximum horizontal displacement of the wall was transferred from the middle and upper part to the bottom of the retaining wall. After 5 years of service, the maximum horizontal displacement corresponding to different geogrid lengths was 14.65–24.08 mm, and with the increase of geogrid length, the horizontal displacement of the retaining wall surface decreased.

After 5, 10, 15 and 20 years of service, the maximum horizontal displacement was 24.08 mm, 27.82 mm, 29.29 mm and 30.26 mm, respectively. It can be seen that the change in horizontal displacement decreased gradually as time went on.

## 6. Conclusions

According to the field measured data on earth pressure, displacement meter and inclinometer conducted 6 years after construction, the mechanical behavior of modular-reinforced earth-retaining wall in use was analyzed, and the performance of the wall was determined to be stable. Furthermore, there was no obvious settlement, indicating that the modular-reinforced earth-retaining wall can be applied to the subgrade of high-speed trains.

(1)  The vertical earth pressure and horizontal earth pressure of the reinforced earth-retaining wall showed a nonlinear distribution along the wall height and increased with the service time. The vertical pressure remained stable after construction and was less than the theoretical value after 6 years. The maximum pressure was obtained at the wall height of 0.35 H.

(2)  Within 6 years of the completion of construction, the strain of each layer of reinforcement basically showed a downward trend with the continuation of time, and the decreasing range was less than 0.88%. The measured value of the potential fracture surface presented a nonlinear distribution. Within the range of 1 m to 5 m, the position of the potential fracture surface was close to the 0.3 H method, while the top and bottom of the retaining wall were far from the theoretical value.

(3)  In the service time of 6 years, the maximum horizontal displacement of the wall was 11.43 mm and the wall remained stable. With the extension of post-construction time, the vertical settlement of the wall top increased gradually, but the settlement rate decreased gradually. Within 6 years after the completion of construction, the maximum settlement value of the wall surface was 46.77 mm. The structure reached a stable state one year after the completion of construction and maintains a good mechanical state and stability during operation.

(4)  During the service period of the retaining wall, the post-construction deformation under different factors gradually decreased as time went on; the variation law of deformation at each time was basically the same and the growth range of the retaining wall structure deformation 5 years after construction was large. The main controlling factors of the horizontal deformation in the service period were the elastic modulus of filler and the reinforcement length.

**Author Contributions:** Writing—original draft preparation, X.L.; writing—review and editing, J.J.; data collection, Y.Z. and Q.Z.; validation, G.Y. and X.W. All authors have read and agreed to the published version of the manuscript.

**Funding:** Natural Science Foundation of Hebei Province (grant No. E2019208159, E2020208071), Doctoral Research Startup Fund of Hebei University of Science and Technology (Grant No. 1181482), Project of Talent Introduction of Hebei Province in 2021 (Key Technologies of Intelligent Construction of New Engineering), Hebei Key Laboratory of Geotechnical Engineering Safety and Deformation Control (Grant No. HWEKF202102), Key Laboratory of Roads and Railway Engineering Safety Control (Shijiazhuang Tiedao University), Ministry of Education (STKF201902), The 333 talent project of Hebei Province (Grant A202001030).

**Institutional Review Board Statement:** Not applicable.

**Informed Consent Statement:** Not applicable.

**Data Availability Statement:** The data used to support the findings of this study are available from the corresponding author upon request.

**Conflicts of Interest:** The authors declare no conflict of interest.

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
