# Peer review of "Performance of Modular-Reinforced Soil-Retaining Walls for an Intercity Railway during Service"

_sustainability, doi:10.3390/su14106084_

Round 1

Reviewer 1 Report

Congratulations to the authors of a very interesting and up-to-date study on the deformation and displacement of a retaining wall from the ground with additional reinforcement.
A very important element of the experiment is the fact that the research lasted 6 years and the obtained results are not a simulation but an actual image of the behavior of the retaining wall.
Please complete the article with detailed information regarding item 3 Field Test. In addition, in my opinion, from point 4, the discussion on d results should be separated from the results. The results are very well presented but I miss a broader explanation of the results in the discussion.
In my opinion, the literature should be supplemented with additional items related to research in this field.

My comments are minor and in my opinion, after reading everything and making some minor changes, I recommend that you publish this work in a journal.

Author Response

Thank you for your suggestion.
1. The detailed information regarding item 3 field test was completed.

2. In point 4, Some discussions and results were separated.

3. The literatures related to research in this field were supplemented.

Reviewer 2 Report

This reinforced soil retaining walls topic is very important for the effecient operation of the important transportation infrastructure; particularly high speed rail and to a lesser extent freeways.

Unfortunately, the Paper's Engish is very poor and there is really no connectivity of the comprehensive field testing/findings to the Conclusions, particularly with respect to tolerable settlements for high speed trains. Exstensive editing of the English language and style is recommended (required). Also: tables and /figures to be captioned/styled properly; Specification referenced; information on tolerable settlements for high speed train operations provided; and then a detailed analysis of the findings completed with connectivity to the Conclusions along with implementation and further research recommendations. 

Author Response

Thank you for your suggestion.
1. The manuscript has been carefully revised by every author with a help from a native speaker.
The written of the paper was updated.

2. The Conclusions part has been modified

3. With a design speed of 250km/h, if the structure was to reach a stable state, its tolerable
settlement should not be greater than 10 cm, and the settlement rate should not be greater than
3 cm per year, according to the Code for Design of High-speed Railway.

Reviewer 3 Report

This is an interesting contribution concerning an instrumented reinforced embankment for high-speed railways. A large amount of data has been measured over six years. In the article, the authors evaluate and comment on the measured data.

I have one comment on the article that is not mentioned and may not be clear to the reader.  What is a subsoil under the embankment? (soil? rock? compressible? )Can't the results of settlement of the top of the embankment be distorted  by settlement of the subsoil?

Author Response

Thank you for your suggestion.
Subsoil under the embankment is lava rock.

Embankment is a reinforced soil structure, it’s a kind of flexible seismic structure with good
deformation resistance, which is coordinated by reinforcement and subsoil.

The reinforced soil embankment depends on the self-weight of the wall system to resist the
instability force caused by the soil and overload on the structure.

To provide embankment stability, it must have sufficient width to prevent sliding at the foundation
and overturning of the subsoil under the lateral soil pressure.

Round 2

Reviewer 2 Report

I have carefully reviewed both the Authors' Comments and the Second Revised Manuscript (ID 1637266). Unfortunately, I still can not recommend publication as an Article in Sustainability. The Second Revised Manuscript has no analysis whatsoever of the field testing results, connectivity, conclusions, or further research recommendations. My review of the First Revised Manuscript was very clear on this: "..  most importantly, there is still no detailed analysis of the findings with connectivity to the Conclusions along with implementation and further research recommendations." The Second Revised Manuscript is now essentially a typical field testing results Report and not an Article. The Abstract should be modified to clearly indicate this.

In brief, the following comments cover my review concerns.

Minor - While editing of English language and style has been completed there is still some minor editing required. 

Moderate - Starting at Line 262 Table 2 is incorrectly referred to (should be Table 3 for this section - noting the Table 3 title should not be Sedimentation..., but rather Settlement,,,).

Major - As indicated above, now essentially a typical field testing results Report and not an Article. 

Author Response

Thank you for your suggestion. The authors have comprehensively revised this paper. The specific modifications are as follows:

  1. The abstract was redescribed, according to the revised paper content.
  2. In the section 1, introduction, the latest research results on the numerical simulation of reinforced earth retaining wall were added.
  3. In section 2 were reintegrated, the project overview and monitoring scheme were combined into remote monitoring in service period, which were described in simple and concise language.
  4. In section 3 of monitoring results and analysis were integrated, and data of vertical earth pressure inside the retaining wall, horizontal lateral earth pressure on the back of wall, reinforcement strain, horizontal displacement and settlement of the retaining wall were deeply analyzed, discussed in detail, and some laws of reference value were obtained.
  5. In section 4 and 5, Numerical simulation was added, in order to study the influencing factors of horizontal deformation in reinforced soil retaining wall. In section 4, the constitutive relationship of the model and material parameters were In section 5 numerical simulation, the effects of filler elastic modulus, reinforcement tensile modulus and reinforcement length on the deformation of the retaining wall were analyzed.
  6. In section 6, conclusions were added the conclusion of numerical simulation.
  7. In addition, some minor comments had also been revised.

In short, with major revision, this paper was clear ideas, thorough analysis and concise conclusions. At the same time, this paper can provide reference for the design of reinforced soil retaining wall.